# The Role of Surgery in the Management of Gastric Cancer: State of the Art

**DOI:** 10.3390/cancers14225542

**Published:** 2022-11-11

**Authors:** Fausto Rosa, Carlo Alberto Schena, Vito Laterza, Giuseppe Quero, Claudio Fiorillo, Antonia Strippoli, Carmelo Pozzo, Valerio Papa, Sergio Alfieri

**Affiliations:** 1Digestive Surgery Unit, Fondazione Policlinico Universitario Agostino Gemelli IRCCS, 00168 Rome, Italy; 2Department of Medical and Surgical Sciences, Università Cattolica del Sacro Cuore, 00168 Rome, Italy; 3Medical Oncology, Fondazione Policlinico Universitario Agostino Gemelli IRCCS, 00168 Rome, Italy

**Keywords:** gastric cancer, early gastric cancer, gastric surgery, gastric lymphadenectomy, extent of gastric surgery, multimodal therapy, gastric cancer neoadjuvant therapy, gastric cancer adjuvant therapy, PIPAC, HIPEC

## Abstract

**Simple Summary:**

Gastric cancer remains the sixth most prevalent malignant tumor worldwide and the third leading cause of cancer-related death. Surgery is the pillar of its treatment at all stages, but the importance of endoscopic treatments and multimodal therapy is growing. The aim of this review is to provide a comprehensive description of the role of surgery for gastric cancer in the modern era.

**Abstract:**

Surgery still represents the mainstay of treatment of all stages of gastric cancer (GC). Surgical resections represent potentially curative options in the case of early GC with a low risk of node metastasis. Sentinel lymph node biopsy and indocyanine green fluorescence are novel techniques which may improve the employment of stomach-sparing procedures, ameliorating quality of life without compromising oncological radicality. Nonetheless, the diffusion of these techniques is limited in Western countries. Conversely, radical gastrectomy with extensive lymphadenectomy and multimodal treatment represents a valid option in the case of advanced GC. Differences between Eastern and Western recommendations still exist, and the optimal multimodal strategy is still a matter of investigation. Recent chemotherapy protocols have made surgery available for patients with oligometastatic disease. In this context, intraperitoneal administration of chemotherapy via HIPEC or PIPAC has emerged as an alternative weapon for patients with peritoneal carcinomatosis. In conclusion, the surgical management of GC is still evolving together with the multimodal strategy. It is mandatory for surgeons to be conscious of the current evolution of the surgical management of GC in the era of multidisciplinary and tailored medicine.

## 1. Introduction

Despite its declining incidence over the last decades, gastric cancer (GC) remains the sixth most prevalent malignant tumor worldwide and the third leading cause of cancer-related death [1]. The most important factors strictly related to this decline are improved refrigeration [2] and effective therapy against Helicobacter pylori [3]. Nevertheless, GC is still burdened with high mortality and its prognosis remains dismal. Indeed, in Western countries, up to two-third of patients are diagnosed with advanced-stage disease (stage III–IV), and even in Eastern countries like Japan, where extensive screening programs exist, this percentage still reaches 50% [1]. Nowadays, a multidisciplinary approach is required for the management of GC. In this context, surgery still represents the mainstay of GC treatment, despite being burdened with significant morbidity (9.1–46%) and mortality (up to 13%), mostly in the case of radical gastrectomy [4]. In the last decades, technological improvements, the employment of innovative chemotherapy regimens, the spread of minimally invasive techniques and multidisciplinary decision-making have paved the way for patient-tailored strategies, with the aim of improving short- and long-term outcomes. The purpose of this narrative review is to provide a comprehensive description of the current surgical management of GC.

## 2. Early Gastric Cancer

Early gastric cancer (EGC) is defined as a cT1 tumor (i.e., limited to the mucosa or submucosa) independently of lymph node involvement [5]. The management of EGC is determined by the risk of lymph node metastasis, which is very low (1–5%) in the case of cT1aN0 differentiated GC, smaller than 2 cm, without ulceration and lymphovascular invasion [6,7]. Tumors with these features could be suitable for curative endoscopic resection, as reported by Eastern and Western authors [8,9,10]. Conversely, a propensity-matched study by Kamarajah et al. [7] showed that endoscopic resection was inferior to surgery regarding long-term survival for cT1aN0 and cT1bN0 GC, but the treatment selection was based only on the clinical T-stage. For these reasons, the Japanese Gastric Cancer Association (JGCA) [11], the European Society of Medical Oncology (ESMO) [12], the National Comprehensive Cancer Network (NCCN) [13], and the Italian Research Group for Gastric Cancer (GIRCG) [14] all currently recommend endoscopic resection as initial treatment for EGC that meets the aforementioned features. In the case of final pathology reporting poorly differentiated patterns, incomplete resection and/or lymphovascular invasion, the endoscopic resection cannot be considered curative due to the high nodal metastases rate (up to 14%) [15], and radical gastrectomy must be performed [11]. Additionally, surgical resection should be considered for gastric cT1b tumors, as the risk of nodal dissemination is 18–32% in these cases [7,16]. In the case of EGC with clinical evidence of lymph node metastases (cN+), patients should be referred to a multidisciplinary discussion for the definition of the appropriate multimodal therapy.

On the other hand, in recent years sentinel lymph node biopsy followed by function-preserving resection has emerged as viable option for EGC not suitable for endoscopic treatment or when curative endoscopic dissection is not achieved. The procedure of node mapping through submucosal injection of indocyanine green or radioactive colloid is routinely employed for breast cancer and melanoma and represents an innovative tool for GC in selected Eastern centers. A radical gastrectomy is warranted in the case of positive lymph nodes, while a stomach-preserving procedure without additional lymphadenectomy may be considered in the case of negative sentinel node biopsy [17]. Stomach-preserving procedures include gastric wedge resections or segmental gastric resections with gastro-gastric anastomosis [18], the latter being associated with better outcomes in terms of quality of life in Eastern centers [19]. To the best of our knowledge, the only trial addressing the long-term survival of organ-sparing gastric resections is the ongoing SENORITA trial [20].

## 3. Surgery for Hereditary Diffuse Gastric Cancer

Hereditary diffuse gastric cancer (HDGC) is an autosomal dominant syndrome associated with a mutation of the CDH1 gene. This syndrome leads to GC with a pathologic pattern characterized by the presence of signet ring cells and multifocal growth [21]. Prophylactic gastrectomy in early adulthood is recommended by the International Gastric Cancer Linkage Consortium (IGCLC) as a standard of care in patients with pathogenic CDH1 mutation associated with a family history of diffuse-type GC [22].

## 4. Non-Metastatic Gastric Cancer

A comprehensive and multimodal approach based on the triad surgery-chemotherapy-radiotherapy is indicated for non-metastatic GC that is more advanced than early GC and in the case of clinical evidence of nodal metastases.

### 4.1. Extent of Gastric Resection

The extent of surgical resection for GC hinges upon tumor size, location, and histological type. As a result, GC surgery may shift from the resection of the whole stomach including the cardia and pylorus, the so-called total gastrectomy, to subtotal gastrectomies. The latter encompasses different technical procedures depending on the resected anatomical region of the stomach:

Distal gastrectomy: up to two-thirds of the distal stomach and pylorus are resected, while the cardia is preserved;

Proximal gastrectomy: the resection includes the cardia and the upper third of the stomach, preserving the antrum and pylorus;

Pylorus-preserving gastrectomy: resection of the central part of the stomach preserving its upper third, the antro-pyloric region enclosing the infra-pyloric vessels and the hepatic branch of the vagus nerve.

According to the JGCA treatment guidelines, both proximal and pylorus-preserving gastrectomy may be considered as therapeutic options only for cT1N0 tumors [11]. Otherwise, the optimal surgical strategy for localized GC greater than EGC (T2–T4a) or with clinically positive lymph nodes (cN+) requires a subtotal or total gastrectomy with a D2 lymphadenectomy. However, the extent and type of surgical resection (subtotal vs. total gastrectomy) remains a subject of debate. The positive impact of organ-sparing surgery for GC on patients’ quality of life and nutritional status has led to the spread of less-demolitive procedures during the last years [23,24,25,26,27,28]. Furthermore, subtotal gastrectomy for distal GC resulted in similar mortality, overall and disease-free survival but less morbidity, compared to total gastrectomy [29,30,31,32,33,34]. The achievement of a complete resection with negative margins is the main goal of oncological gastric surgery, as it still plays the role of the only potentially curative intervention for GC. Specifically, the concept of R0 refers to a microscopically margin-negative resection with no evidence of macroscopic or microscopic tumor cells in the primary tumor site [35]. Distal gastrectomy is generally the privileged option for distal GC, strictly when an adequate proximal resection margin can be achieved [11,12,13]. Otherwise, total gastrectomy is mandatory. Currently, there is a considerable heterogeneity across Western and Eastern cancer guidelines on the proper distance between GC and resection margins to achieve R0 radical surgery (Table 1).

Interestingly, only the JGCA and the GIRCG guidelines conceive less-extensive gastric resections tailored on the histopathological growth pattern of GC [11,14]. Despite the aforementioned guidelines, several authors highlighted the absence of any significant influence of proximal margin length on GC overall survival [36,37,38,39], unlike others which advocated for a patient-tailored proximal margin length as an independent prognostic factor [40]. Recently, Maspero et al. assessed the impact of JGCA-recommended resection margins on long-term outcomes of 279 consecutive Western patients [41]. Analyzing 220 distal gastrectomies, the authors found that application of JGCA guidelines resulted in improved overall survival and a more organ-preserving surgical strategy, owing to a significant lower number of total gastrectomies compared with NCCN and ESMO guidelines (30% vs. 31% vs. 47%, respectively) without compromising oncological outcomes [41].

### 4.2. Lymphadenectomy

The extent of lymphadenectomy in radical gastrectomy has long been a source of controversy. The 8th International Union for Cancer Control/American Joint Committee on Cancer (IUCC/AJCC) classification recommends the analysis of at least 16 retrieved nodes for reliable staging [42,43]. Regional lymph nodes for GC were divided into 16 stations according to the JGCA classification [44,45] (Table 2).

Lymphadenectomy for GC is classified as follows:

D1 lymphadenectomy presupposes the removal of the perigastric lymph nodes;

D2 lymphadenectomy entails the resection of perigastric lymph nodes plus those along the left gastric, common hepatic and splenic arteries and the coeliac trunk.

In addition, the JGCA guidelines introduced the concept of D1+ lymphadenectomy as a node dissection more extended than D1 but not fulfilling the criteria of a D2 level [11]. Japanese surgeons considered D1+ lymphadenectomy for T1a tumors not suitable for endoscopic resection and for those of undifferentiated-type or greater than 1.5 cm T1b GC [11]. The NCCN, the ESMO, the JGCA and the GIRCG guidelines currently recommend D2 lymphadenectomy as the gold standard procedure for potentially curable cT2–T4a tumors and for cN+ tumors [11,12,13,14]. Additionally, only the Japanese Guidelines proposed a surgery-related and cancer-tailored technique of D2 lymphadenectomy. Indeed, the removal of distal splenic artery lymph nodes (JGCA node station 11d) is indicated for total gastrectomy, but not for distal gastrectomy, to fulfill the criteria of D2 level [11]. D2 lymphadenectomy, a technically more challenging procedure than D1 lymphadenectomy, represents the standard of care in Eastern countries [11], while it is performed only in experienced high-volume centers and medically fit patients in Western countries [12,13]. Experience from Japanese and Korean studies revealed excellent survivals associated with a lymph node dissection extended to the second D level [46,47,48,49]. Otherwise, several European randomized controlled trials comparing oncological outcomes of D2 versus D1 lymphadenectomy failed to prove any five-year survival superiority related to extensive node dissection [50,51,52,53]. The results of the Dutch Gastric Cancer Group Trial demonstrated a higher postoperative morbidity and mortality in the D2 group and comparable five-year overall survival [52]. Similar results were reported by the Italian Gastric Cancer Study Group (IGCSG) [53] and the Medical Research Council trial [50], despite the high rate of splenectomy, and distal pancreatectomy in D2 gastrectomy in the UK study should be considered as a confounding factor [51]. The 1five-year follow-up version of the Dutch trial showed a statistically significant difference in term of GC-related death (37% in D2 group vs. 48% in D1 group) and locoregional recurrence (12% in D2 group vs. 22% in D1 group) and even a not significantly higher overall survival (29% in D2 group vs. 21% in D1 group; *p* = 0.34) [54]. Moreover, after the exclusion of the subgroup of pancreatectomies and splenectomies, the 15-year overall survival increased to 32% in the D2 group (vs. 22% in D1 group; *p* = 0.006) [54]. Recently, the updated IGCSG experience also highlighted a significative improvement of disease-specific survival (51.4% in D2 group vs. 29.4% in D1 group) and GC-related mortality (43% in D2 group vs. 65% in D1 group) after a 15-year period in patients with advanced GC (pT > 1) and lymph node metastases who underwent D2 lymphadenectomy [55]. In conclusion, D2 lymphadenectomy is currently indicated in advanced GC [11,12,13] but in Western countries it should be performed only in selected patients by experienced surgeons in referral centers [12,13].

### 4.3. Minimally Invasive Procedures

The feasibility and safety of minimally invasive gastric surgery was first reported by Kitano et al. in 1995 [56]. Since then, laparoscopic gastrectomy has arisen as a non-inferior technique to conventional open gastrectomy [57,58] and its effectiveness has been analyzed by several randomized clinical trials (Table 3).

According to the results of a recent systematic review and meta-analysis, laparoscopic distal gastrectomy had less perioperative blood loss, fewer postoperative complication, faster recovery of bowel function, equivalent oncological surgical precision (defined by lymph node yield, resection margins and anastomotic leakage) compared with open surgery, at the cost of longer operative time [85]. Nevertheless, the technical complexity of laparoscopic gastric surgery is reflected by its steep learning curve, requiring 20 to 40 cases and up to 100 cases for laparoscopic distal and total gastrectomy, respectively, to achieve proficiency [86,87].

During the last years, robotic surgery has emerged as a new minimally invasive technique for oncological gastric surgery. Indeed, the robotic platform has overcome the limitations of laparoscopy through the introduction of several advantages, such as the wider range of motions and surgical dexterity, 3D high-definition view, tremor reduction, better ergonomics and a faster learning curve [88]. Despite higher costs and longer operative time than laparoscopy, robotic gastrectomy leads to similar or slightly improved complication rates and lymph node harvest [89,90,91,92,93], and comparable three-year overall and relapse-free survival [94], resulting in advantages also for patients with visceral fat obesity [95]. The benefits derived from the robotic platform were more conspicuous when considering only robotic distal gastrectomy (RDG) [96]. Recently, the safety and efficacy of minimally invasive distal gastrectomy were evaluated in a randomized controlled trial comparing robotic versus laparoscopy [97]. The robotic approach was then associated with faster recovery, lower postoperative complications and inflammatory response, higher dissection rates of extra/perigastric lymph nodes and shorter delay of adjuvant chemotherapy [97]. The higher effectiveness and safety of RDG in terms of surgical and oncological outcomes was ultimately confirmed by the Gong et al. meta-analysis [98].

### 4.4. Multimodal Treatment

All patients with a clinical T2 or more advanced disease (i.e., clinically positive lymph nodes) should be discussed in a multidisciplinary context to define the appropriate multimodal treatment [99]. Providing a comprehensive description of the pharmacological treatment of GC is far beyond the purpose of this review. However, given the positive impact of multimodal strategies in the treatment of locally advanced GC becoming clearer over time, it is mandatory to describe the efforts to improve the results obtained with surgery alone, including neoadjuvant and adjuvant strategies.

### 4.5. Neoadjuvant/Perioperative Therapy

According to the latest version of NCCN Guidelines [13], two main strategies are recommended as alternatives to upfront surgery for cT2 or higher locoregional disease, namely perioperative chemotherapy and preoperative chemoradiation. These recommendations derive from the results of several clinical trials which have directly compared surgery alone with neoadjuvant or perioperative chemotherapy, demonstrating a survival benefit for the multimodal approaches. Initially, the Medical Research Council Adjuvant Gastric Infusional Chemotherapy (MAGIC) trial [100] randomly assigned 503 patients with potentially resectable gastric, distal esophageal or esophagogastric junction adenocarcinomas to upfront surgery or preoperative (three neoadjuvant plus three adjuvant cycles) chemotherapy with epirubicin, cisplatin and infusional fluorouracile (ECF), showing better overall and progression-free survival for patients with locoregional disease (cT2 or higher or cN > 0) who received perioperative chemotherapy. Notably, only 42% of these patients were able to complete the whole protocol treatment. Subsequently, the phase II/III FLOT4-AIO (Fluorouracil, Leucovorin, Oxaliplatin, and Docetaxel 4—Arbeitsgemeinschaft Internistische Onkologie) trial [101] demonstrated a better overall survival and three-year overall survival for patients with gastric or gastroesophageal cancer who received FLOT regimen (four preoperative and four post-operative Fluorouracil, Leucovorin, Oxaliplatin, and Docetaxel cycles) as compared to patients who received epirubicin-based therapy (ECF or ECX—epirubicin, cisplatin and capecitabine). These results made FLOT regimen the first choice for patients with excellent performance status. However, due to the considerable toxicity associated with FLOT, ECF is still recommended in selected patients with good performance status. Finally, for patients who have good to moderate performance status, the suggested perioperative regimen is fluorouracil and oxaliplatin (FOLFOX) or capecitabine and oxaliplatin (CAPOX) [13]. Ongoing trials are testing the addition of immune checkpoint inhibitors to the FLOT regimen for mismatch repair-deficient tumors that express programmed death-ligand 1, such as pembrolizumab in the KEYNOTE-585 trial [102], etezolizumab in the DANTE study [103], and durvalumab [104]. Moreover, ongoing studies are investigating the addition of trastuzumab to a current chemotherapy regimen in a perioperative setting for patients with ERBB2 (HER2 or HER2/neu)–positive tumors [105]. With regards to the use of chemoradiotherapy in the neoadjuvant setting, several small studies have demonstrated the possibility to produce a pathologic response in resectable GC [106,107,108,109]. Nonetheless, its clinical employment is still questionable and under investigation due to the lack of randomized controlled trials demonstrating a survival benefit in noncardia GC. Indeed, the only available trials analyzed esophageal, gastroesophageal junction (GEJ) and gastric cardia tumors only [110,111]. The answer to whether preoperative chemoradiotherapy is better than upfront surgery alone, neoadjuvant chemotherapy or adjuvant therapy after surgery will probably come from several ongoing trials, such as the Chemoradiotherapy after Induction Chemotherapy in Cancer of the Stomach II (CRITICS-II) [112], the ESOPEC trial [113] and the Trial of Preoperative Therapy for Gastric and Esophagogastric Junction Adenocarcinoma (TOPGEAR) trial [114].

### 4.6. Adjuvant Therapy

For patients who have undergone upfront gastric resection without receiving any neoadjuvant therapy, the current NCCN guidelines recommend adjuvant treatment rather than surgery alone in patients with pT3-4 and/or N + GC [13]. Options for adjuvant therapy include chemotherapy alone and chemoradiotherapy plus chemotherapy. In particular, postoperative chemotherapy with capecitabine and oxaliplatin or fluorouracil and oxaliplatin is indicated for patients who have undergone gastric resection with D2 lymph node dissection; postoperative chemoradiation plus chemotherapy (fluoropyrimidine before and after fluoropyrimidine-based chemoradiation) is indicated for patients who received less than D2 lymph node dissection; chemoradiotherapy alone for patients who received R1-R2 gastrectomy. Finally, for selected patients with high-risk features (including poorly differentiated or higher-grade cancer; lymphovascular invasion; neural invasion; <50 years of age; lymph node dissection less than D2), the NCCN guidelines recommend chemoradiation plus chemotherapy even in the case of pT2, N0 disease. With regards to chemotherapy, the optimal regime has not been established yet: data deriving from several phase III trials indicate CAPOX, FOLFOX and S-1 with or without docetaxel as the main alternatives. These indications derive from the results of several key randomized clinical trials, shown in Table 4.

Interestingly, a retrospective study based on data from the National Cancer Database [126] found a benefit for chemoradiotherapy with regard to five-year overall survival in patients with positive lymph nodes and lymphovascular invasion, but not in patients without lymphovascular invasion, suggesting the need for tailored approaches for selected groups of patients.

## 5. Metastatic Gastric Cancer

While chemotherapy still represents the mainstay of treatment for patients with metastatic gastric cancer, recent advances in systemic therapy are expanding the surgical indications even to patients with stage IV gastric cancer, leading to a more tailored approach.

### 5.1. Resectable Metastatic Disease

Several small retrospective studies analyzing the outcome of surgery in patients with limited metastatic disease have shown a possible survival benefit [127,128,129,130]. However, they are often characterized by an important selection bias, as the patients undergoing surgery often have better prognosis and a smaller disease burden than those who received gastrojejunostomy or no surgery at all. Moreover, many factors that influence survival, such as systemic chemotherapy, are not always considered. Resection of hepatic metastasis has been rarely performed in patients with isolated gastric cancer liver secondary lesions [131,132], and long-term survival in these patients represents a rare event [131,132,133,134,135]. For these reasons, no consensus regarding appropriate patient selection criteria currently exists. Pulmonary metastasectomy may also be taken into consideration for highly selected patients, potentially leading to long-term survival, but data are still insufficient [136,137,138,139]. However, encouraging results regarding potential long-term survival in selected patients with M+ gastric cancer after modern chemotherapy schemes followed by extensive surgery come from new and ongoing trials. The Arbeitsgemeinschaft Internistische Onkologie—Fluorouracil, Leucovorin, Oxaliplatin, and Docetaxel 3 (AIO-FLOT3) phase II trial [140] investigated the outcomes of perioperative FLOT plus surgery in limited metastatic disease (one metastatic organ site with or without retroperitoneal nodes) suitable for R0 and macroscopically complete (R1) resection of the primary tumor and metastatic lesions at restaging, respectively. The authors found a higher median overall survival in patients who underwent surgery than in patients who only received FLOT (31 vs. 16 months, respectively). The results of the follow-up RENAISSANCE (AIO-FLOT5) phase III trial [141], which is currently randomizing patients without progression after four cycles of FLOT to additional FLOT or surgery followed by postoperative FLOT, are awaited. Similar results were obtained from a large multicenter retrospective cohort study, namely the Conversion Therapy for Stage IV Gastric Cancer 1 (CONVO-GC-1) trial [142]. The authors included 1206 patients who received chemotherapy followed by radical surgical resection, reporting 56.6 months (95% CI, 46.4–74.5) of median overall survival for patients who underwent R0 resection, as compared to 36.7 months (95% CI, 34.4–40.0) for all patients.

### 5.2. Peritoneal Disease

After liver, the peritoneum represents the second most common site of gastric cancer metastatic spread [143].

#### 5.2.1. Hyperthermic Intraperitoneal Chemotherapy (HIPEC)

For patients with isolated peritoneal metastasis, the role of cytoreductive surgery (CRS) with or without hyperthermic intraperitoneal chemotherapy (HIPEC; normally employing mitomycin C at 40 °C to 43 °C) still remains controversial and is not unanimously considered as the standard approach [144,145,146,147]. Indeed, trials comparing CRS with or without HIPEC and systemic chemotherapy are still lacking, and protocols vary largely by institution. Currently, the reported median overall survival after CRS plus HIPEC ranges from 11 to 19 months, and one of the main prognostic factors is represented by the quality and completeness of surgical cytoreduction [144,145]. The main evidence of improved survival derives from a phase III trial from China [145] and the CYTO-CHIP French study [144]. In particular, the latter analyzed data from nineteen French cancer treatment centers, investigating the impact of CRS alone as compared to CRS plus HIPEC in selected patients with isolated peritoneal metastases from gastric cancer. The authors found better median survival in the group of patients receiving HIPEC after CRS (19 vs. 12 months) and better five-year recurrence-free survival as well (17% vs. 4%). While demonstrating statistical significance, these results should not be overinterpreted, as less than one patient per center per year was recruited, which in fact represents a very highly selected population. Moreover, a significant percentage of patients had only positive cytology. With regard to patients with occult or minimal-volume peritoneal disease from GC (i.e., positive cytology only or radiographically occult peritoneal disease), occasional long-term survival has been reported in the case of curative-intent surgery following negative cytology achieved after neoadjuvant chemotherapy [148,149]. Moreover, a small phase II study from the US-tested HIPEC following systemic chemotherapy in 19 patients diagnosed with minimal peritoneal carcinomatosis showed 30.2 months of median overall survival [150]. Remarkably, some studies have shown that surgery itself can become a vehicle of iatrogenic peritoneal dissemination, especially through division of lymphatic channels in patients with high T and N disease [151,152]. In this context, the ongoing PILGRIM HIPEC-01 trial is currently testing this hypothesis, randomizing patients with T3–T4 Nx M0 gastric cancer undergoing curative-intent surgery to adjuvant chemotherapy or adjuvant HIPEC followed by adjuvant chemotherapy. Despite preliminary results showing a favorable safety profile for HIPEC, definitive results data are still awaited [153].

#### 5.2.2. Pressurized Intraperitoneal Chemotherapy (PIPAC)

In the setting of unresectable peritoneal metastasis and malignant ascites from GC, a novel treatment using aerosolized chemotherapy, namely pressurized intraperitoneal chemotherapy (PIPAC), has been increasingly used in the last years with the goal of increasing patients’ survival. Reymond et al. first reported the significant impact of PIPAC on tumor response with a 25% complete pathological response [154]. Alyami et al. tested the outcomes of cisplatin and doxorubicin via PIPAC in 42 patients with unresectable peritoneal disease, showing a median overall survival of 19.1 months and only 6.1% morbidity [155]. Another study by Di Giorgio and colleagues reported a pathological response in 61.5% of 28 consecutive patients, with a median overall survival of 12.3 months for the entire cohort. Interestingly, the authors reported an overall survival of 15 months in the subgroup of patients undergoing more than one PIPAC procedure [156]. Several other studies have shown the safety and the effectiveness of PIPAC with cisplatin (7.5 mg/m^2^) and doxorubicin (1.5 mg/m^2^) in patients with unresectable peritoneal metastasis from GC [155,157,158]. However, definitive data regarding the safety and efficacy of PIPAC, as well as the optimal drugs and dose to be used, are still awaited. The PIPAC EstoK 01 [159] is an ongoing prospective randomized multicenter phase II trial assigning patients with peritoneal dissemination from gastric cancer to three cycles of PIPAC with oxaliplatin plus systemic intravenous chemotherapy or systemic chemotherapy alone. Two ongoing studies concerning the use of oxaliplatin PIPAC are aiming to find its optimal dose [144,160]. Finally, the PIPAC-GA01 trial is currently evaluating the safety and efficacy of PIPAC with three single doses of doxorubicin and cisplatin in patients with recurrent gastric cancer [161].

## 6. Palliation

The majority of patients diagnosed with GC will require palliative treatment due to the dismal prognosis of the disease [162]. While chemotherapy represents the most effective treatment for patients with metastatic disease, palliation of symptoms such as obstruction, bleeding or perforation may require multidisciplinary management employing surgery, endoscopy, radiotherapy, or other approaches.

### 6.1. Surgical Palliation

For patients diagnosed with advanced GC, palliative gastrectomy plays a role in rapidly relieving symptoms such as obstruction, perforation, bleeding, pain, and nausea, but did not show any survival benefit. Indeed, the Japan Clinical Oncology Group 0705 and Korean Gastric Cancer Association 01 (REGATTA) phase III trial [163] tested the survival benefit of gastrectomy in 175 patients with advanced GC and a single non-curable factor (such as peritoneum, liver or para-aortic metastasis), who were randomly assigned to S-1 plus cisplatin chemotherapy alone or to gastrectomy followed by the same systemic treatment. The study did not show any improvement in terms of overall survival, while increasing the incidence of serious adverse events related to chemotherapy. Palliative gastrojejunostomy also plays a role in symptom relief, especially when other nonsurgical methods cannot be used. Moreover, when performed laparoscopically, it can lead to less blood loss and shorter length of hospital stay than an open approach, as shown by several case reports and small studies [164]. Among surgical options, gastrectomy and gastrojejunostomy have not yet shown a significant difference in outcomes according to available data [165].

### 6.2. Non-Surgical Palliation

External beam radiation therapy plays a role in the control of obstruction, pain and especially bleeding in patients with unresectable gastric cancer [165,166,167,168,169,170,171,172], with a good tolerance of the treatment and a low rate of toxicity. While no controlled studies comparing radiotherapy with endoscopic or surgical treatments for symptomatic palliation currently exist, it is evident that the late onset of the effect of radiotherapy makes it not the ideal strategy for all the symptoms. In particular, in the case of obstruction, endoscopic stent placement represents a faster and less invasive alternative to surgery and radiotherapy. While not being inferior to palliative gastrojejunostomy in terms of efficacy and complications, it seems to lead to faster relief of obstructive symptoms and shorter length of hospital stay. However, it remains particularly indicated for patients with short life expectancy, due to a higher reintervention rate and a lower durability as compared to surgical palliation [173]. Additionally, endoscopy has a role in the relief of dysphagia through endoscopic laser treatment (especially in patients with GEJ tumors) [174,175,176], and in the treatment of hemorrhage through laser photocoagulation (in particular for large diffusely bleeding tumors) [177,178], argon plasma coagulation and application of hemostatic nanopowder [179].

## 7. Conclusions

Surgery still represents the mainstay of treatment for GC and its effectiveness has been enhanced by the introduction of novel minimally invasive surgical techniques and systemic therapies. The Eastern experience in the management of EGC underlined the importance of preserving quality of life without jeopardizing oncologic radicality, through the employment of endoscopic resection, sentinel lymph node biopsy and organ-sparing procedures. On the other side, more advanced non-metastatic GCs require a comprehensive and multimodal approach based on the surgery-chemotherapy-radiotherapy triad. Nevertheless, the lack of a global consensus on the optimal surgical strategy for GC has led to many discrepancies between Western and Eastern guidelines, mainly regarding the extent of surgical resection and lymphadenectomy. For this reason, a lymphadenectomy extended to the D2 level represents the standard of care in Eastern centers, while it is performed only in high-volume Western institutions for medically fit patients. In recent years, the spread of novel chemotherapy delivery topical systems, such as HIPEC and PIPAC, has changed the paradigm of end-stage GC with peritoneal dissemination. Similarly, the surgical management of metastatic GC disease has been allowed by the improvement of chemoradiotherapy regimens. Finally, the molecular classification of GC has recently been published, paving the way for new perspectives of tailored treatment options [180,181,182]. However, to date, its surgical implications are only speculative [183] and the integration in clinical practice is still awaited.

## Figures and Tables

**Table 1 cancers-14-05542-t001:** Recommendations on the extent of resection for GC.

Cancer Society Guidelines	Recommendation
National Comprehensive Cancer Network [13]	The minimum margin length is not specified
European Society of Medical Oncology [12]	Subtotal gastrectomy should be performed with a minimum margin length of at least 5 cm for intestinal-type GC and at least 8 cm for diffuse-type GC
Japanese Gastric Cancer Association [11]	Subtotal gastrectomy should be performed with a minimum margin length of at least 2 cm for T1 tumors, 3 cm for T2 or deeper tumors with an expansive growth pattern and 5 cm for T2 or deeper tumors with an infiltrative growth pattern
Italian Research Group for Gastric Cancer Guidelines [14]	Subtotal gastrectomy should be performed with a minimum margin length of at least 2 cm for T1 tumors, 3 cm for T2 or deeper tumors with an expansive growth pattern and 5 cm for T2 or deeper tumors with an infiltrative growth pattern and diffuse Lauren histotype

**Table 2 cancers-14-05542-t002:** Japanese Gastric Cancer Association Lymph Node Stations.

Stations	Anatomical Location
1–6	Perigastric
	1: Right of the cardia
	2: Left of the cardia
	3a: Lesser curvature (branches of the left gastric artery)
	3b: Lesser curvature (2nd and distal branches of the right gastric artery)
	4sa: Greater curvature (short gastric arteries)
	4sb: Greater curvature (left gastroepiploic artery)
	4d: Greater curvature (2nd and distal branches of the right gastroepiploic artery)
	5: Superior to the pylorus
	6: Inferior to the pylorus
7	Left gastric artery
8	Common hepatic artery
	8a: Anterior
	8p: Posterior
9	Coeliac axis
10	Splenic hilum
11	Splenic artery
	11p: Proximal
	11d: Distal
12	Hepatoduodenal ligament
	12a: Proper hepatic artery
	12b: Common bile duct
	12p: Portal vein
13	Posterior to the pancreas head
14	Superior mesenteric vein
15	Middle colic vein
16	Para-aortic
	16a1: Hiatus
	16a2: Between celiac artery and left renal vein
	16b1: Between left renal vein and inferior mesenteric artery

Sourced from [44].

**Table 3 cancers-14-05542-t003:** Randomized clinical trials on laparoscopic versus open gastrectomy for gastric cancer.

Authors	Patients	Procedures	Results
Kitano et al., 2002 [59]	28 EGC	Distal gastrectomy	Faster recovery, less pain, and less compromised pulmonary function in the LPS group
Fujii et al., 2003 [60]	20 EGC	Distal gastrectomy	Better preservation of Th1 immune response in the LPS group
Huscher et al., 2005 [61]	59 T1-4 and N0-2 GC	Distal gastrectomy	No difference in terms of mean number of resected lymph nodes, mortality, morbidity, five-year OS and DFS. LPS was associated with lower intraoperative blood loss, earlier resumption of oral intake, and earlier hospital discharge.
Hayashi et al., 2005 [62]	28 EGC	Distal gastrectomy	No difference in terms of oncological radicality. Shorter postoperative epidural anesthesia, lower IL-6 and CRP levels, without major postoperative complications in the LPS group.
Lee et al., 2005 [63]	47 EGC	Distal gastrectomy	Similar oncological outcomes, but fewer pulmonary complications in the LPS group
Kim et al., 2008 [64]	164 EGC	Distal gastrectomy	LPS-related advantages regarding QoL, intraoperative blood loss, analgesic use, and postoperative hospital stay.
Kim et al., 2010 [65]	342 EGC	Distal gastrectomy	No significant difference in morbidity and mortality rate.
Kim et al., 2013 [66]	164 EGC	Distal gastrectomy	Similar overall postoperative morbidity, QoL, five-year OS and DFS. Mild complications were lower in the LPS group.
Sakuramoto et al., 2013 [67]	64 EGC	Distal gastrectomy	LPS resulted in less postoperative pain with similar short-term outcomes than open surgery.
Takiguchi et al., 2013 [68]	40 EGC	Distal gastrectomy	Benefits related to LPS were faster recovery, less intraoperative blood loss, less postoperative pain, smaller wound size, shorter postoperative hospital stay, and better levels of CRP and SaO2.
Hu et al., 2015 [69]	66 stage I-III GC	Distal gastrectomy	LPS was associated with lower morbidity, less intraoperative blood loss, shorter hospital stay, faster recovery and better humoral and cellular immune response.
Hu et al., 2016 [70] CLASS-01 Trial	1056 T2-4a and N0-3 GC	Distal gastrectomy with D2 lymphadenectomy	Similar node-dissection compliance, morbidity and mortality rate.
Kim et al., 2016 KLASS-01 Trial	1416 EGC	Distal gastrectomy with D1+ lymphadenectomy	LPS resulted in lower wound complication rate, comparable overall morbidity and mortality.
Yamashita et al., 2016 [71]	63 EGC	Distal gastrectomy	LPS was associated with less long-term wound pain.
Katai et al., 2017 [72]	921 T1-2 and N0-1 GC non-endoscopically suitable	Distal gastrectomy	LPS was safe as open surgery presenting similar short-term clinical outcomes, with a significantly higher operative time but smaller blood loss.
Park et al., 2018 [73] COACT 1001 trial	204 T2-4a and N0-2 GC	Distal gastrectomy with D2 lymphadenectomy	No significant differences in three-year DFS, morbidity and overall lymphadenectomy noncompliance rate, despite the latter being significantly higher for stage III GC in the LPS group.
Shi et al., 2018 [74]	328 T2-3 and N0-3 GC	Proximal, distal and total gastrectomy with D2 lymphadenectomy	LPS resulted to be safe and feasible procedure in locally advanced GC compared to open surgery.
Shi et al., 2019 [75]	328 T2-3 and N0-3 GC	Proximal, distal and total gastrectomy with D2 lymphadenectomy	No difference in terms of five-year OS, DFS and recurrence rate.
Wang et al., 2019 [76]	446 T2-4a and N0-3	Distal gastrectomy with D2 lymphadenectomy	No difference in terms of 30-day morbidity and mortality, three-year DFS and in compliance rate of D2 lymph node dissection.
Li et al., 2019 [77]	96 T2-4a and N+ GC after neoadjuvant therapy	Distal gastrectomy with D2 lymphadenectomy	LPG gastrectomy resulted in a lower overall complication rate, less pain, similar postoperative recovery, better adjuvant chemotherapy completion rate and comparable mortality.
Lee et al., 2019 [78] KLASS-02 Trial	1050 T2-4a and N0-1 GC	Distal gastrectomy with D2 lymphadenectomy	LPS was significantly associated with faster recovery, lower early morbidity rate, postoperative pain and analgesic use, and shorter hospital stay with no difference in terms of mortality and totally retrieved lymph nodes.
Yu et al., 2019 [79] CLASS-01 Trial	1056 T2-4a and N0-3 GC	Distal gastrectomy with D2 lymphadenectomy	No difference in terms of three-year DFS.
Liu et al., 2020 [80] CLASS-02 trial	227 T1-2 and N0-1 (stage I) GC	Total gastrectomy	No difference in terms of overall postoperative complication rate and mortality.
Hyung et al., 2020 [81] KLASS-02 Trial	1050 T2-4a and N0-1 GC	Distal gastrectomy with D2 lymphadenectomy	No difference in terms of three-year relapse-free survival rate.
Van de Veen et al., 2021 [82] LOGICA Trial	227 T1-4a and N0-3b GC	Total or distal gastrectomy with D2 lymphadenectomy	No difference in terms of postoperative complications, in-hospital mortality, 30-day readmission rate, R0 resections, median lymph node harvested, one-year OS, and one-year global health-related QoL.
Huang et al., 2022 [83] CLASS-01 Trial	1056 T2-4a and N0-3 GC	Distal gastrectomy with D2 lymphadenectomy	Similar five-year OS.
Son et al., 2022 [84] KLASS-02 Trial	1050 T2-4a and N0-1 GC	Distal gastrectomy with D2 lymphadenectomy	Five-year OS and relapse-free survival rates were not significantly different between LPS and open surgery.

EGC, Early Gastric Cancer; LPS, Laparoscopic; GC, Gastric Cancer; OS, Overall Survival; DFS, Disease-Free Survival; QoL, Quality of Life; IL-6, Interleukin-6; CRP, C-Reactive Protein; CLASS-01, Chinese Laparoscopic Gastrointestinal Surgery Study 01; KLASS-01, Korean Laparoendoscopic Gastrointestinal Surgery Study 01; KLASS-02, Korean Laparoendoscopic Gastrointestinal Surgery Study 02; CLASS-02, Chinese Laparoscopic Gastrointestinal Surgery Study 02; LOGICA, Laparoscopic versus Open Gastrectomy for gastrIc Cancer.

**Table 4 cancers-14-05542-t004:** Key trials on adjuvant chemotherapy and chemoradiotherapy for gastric cancer.

Authors	Patients	Groups	Results
Adjuvant Chemotherapy			
Zhang et al., 2011 [115] (China)	80 resected (R0) GC	38: 5-FU/LV; 42: 5-FU/LV + oxaliplatin	Improved RFS and OS for FOLFOX regime: 3-year RFS: 30 vs. 16 months, *p* < 0.05; 3-year OS: 36 vs. 28 months, *p* < 0.05
Sasako et al., 2011 [116] (Japan) ACTS-GC trial	1034 Stage II–III resected (D2) GC	515: S-1; 519: observation	Improved OS for adjuvant S-1: 5-year OS 71.7% vs. 61.1% (HR 0.669; 95% CI, 0.540–0.828).
Bang et al., 2012–2014 [117,118] (China, Taiwan, South Korea) CLASSIC trial	1035 Stage II-IIIB resected (D2) GC	520: CAPOX; 515: observation	Improved DFS for adjuvant CAPOX: 3-year DFS 74% vs. 59% (HR 0.56, 95% CI 0.44–0.72; *p* < 0.0001); 5-year DFS 68% vs. 53% (HR 0.66, 95% CI 0.51–0.85; *p* = 0.0015) Improved OS for adjuvant CAPOX: 78% vs. 69%
Yoshida et al., 2019 [119] (Japan) Interim analysis of JACCRO GC-07 trial	913 Stage III resected (R0) GC	454: S-1 + docetaxel; 459: S-1	Improved ReFS for S-1 plus docetaxel: 3-year ReFS 66% vs. 50% (HR 0.632; 99.99% CI, 0.400 to 0.998; *p* < 0.001)
**Adjuvant chemotherapy plus chemoradiotherapy**			
Macdonald et al., 2001 [120] (US) INT 0116	556 resected gastric/GEJ cancer	281: CRT + 5-FU/LV; 275: observation	Improved OS with CRT: 36 vs. 27 months (HR 1.35; 95% CI 1.09–1.66; *p* < 0.001)
Dikken et al., 2010 [121] (The Netherlands)	91 resected GC	5: CRT + LV 39: CRT + X 47: CRT + XP 694: surgery only (from the DGCT trial)	Fewer local recurrences after CRT: 2% vs. 8%; *p* = 0.001
Yu et al., 2012 [122] (China)	68 T3/T4 and/or N+ resected GC	34: 5-FU/LV + CRT; 34: 5-FU/LV	Improved OS and DFS for CRT: 3-year OS 67.7%, vs. 44.1 (*p* < 0.05); 3-year DFS 55.8 vs. 29.4% (*p* < 0.05)
Park et al., 2015 [123] (South Korea) ARTIST trial	458 resected (D2) GC	230: XP + CRT; 228: XP	DFS not different: HR 0.74; 95% CI 0.52 to 1.05; *p* = 0.0922 OS not different: HR 1.130; 95% CI 0.775 to 1.647; *p* = 0.5272)
Cats et al., 2018 [124] (The Netherlands) CRITICS	788 Stage IB-IVA resectable gastric/GEJ cancer	395: ECX/EOX + surgery + CRT; 393: ECX/EOX + surgery + ECX/EOX	OS not different: 37 vs. 43 months (HR 1.01; 95% CI 0.84–1.22; *p* = 0.90)
Park et al., 2021 [125] (South Korea) ARTIST-II	546 Stage II-III pN > 0 resected (D2) GC	183: SOX + CRT; 181: SOX; 182: S-1	DFS not different between SOX and SOX + CRT; 3-year DFS: 72.8% SOX + CRT vs. 74.3% SOX vs. 64.8% S-1 (HR 0.97; 0.66–1.42; *p* = 0.879)

GC, Gastric Cancer; CLASSIC, Capecitabine and Oxaliplatin Adjuvant Study in Stomach Cancer; CAPOX, capecitabine and oxaliplatin; LV, Leucovorin; FOLFOX, Folinic acid, fluorouracil and oxaliplatin; DFS, Disease free survival; HR, Hazard ratio; OS, Overall survival; RFS, Recurrence-free survival; ACTS-GC, Adjuvant Chemotherapy Trial of TS-1 for Gastric Cancer; 5-FU, 5-Fluorouracil; JACCRO, Japan Clinical Cancer Research Organization; ReFS, Relapse-free survival; GEJ, Gastroesophageal junction; CRT, Chemioradiotherapy; ARTIST, Adjuvant Chemoradiation Therapy in Stomach Cancer; XP, Capecitabine and cisplatin; SOX, S-1 and oxaliplatin; CRITICS, Chemoradiotherapy After Induction Chemotherapy in Cancer of the Stomach; ECX, epirubicin, cisplatin, and capecitabine; EOX, epirubicin, oxaliplatin, and capecitabine; DGCT, Dutch Gastric Cancer Group Trial; T, tumor stage; N, nodal stage from TNM staging system, as per the American Joint Committee on Cancer, 8th Edition.

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
