# Peer review of "The Role of Surgery in the Management of Gastric Cancer: State of the Art"

_cancers, 2022, doi:10.3390/cancers14225542_

Round 1
Reviewer 1 Report
The manuscript is an extensive review of the surgical management of gastric cancer. Surgery is a fundamental treatment in all stages of gastric cancer. To do this, they specify the procedures according to the stage of the cancer. It is a comprehensive review, well written and organized. Some of the tables should be placed in landscape format.
Author Response
We thank the reviewer for the time spent to analyze the manuscript and providing this useful suggestion. The format of Table 3 was modified as requested.
Reviewer 2 Report
Thank you for giving me an opportunity to evaluate this review article entitled “The role of surgery in the management of gastric cancer: state of the art”. This review article is very educational and well organized based on the latest findings. Some minor concerns should be addressed before publication.
Points
1. The study design of each study had better be added to the Table 3.
2. (Page 10, line 307) Chemotherapy therapy → Chemotherapy
Author Response
We would like to thank the Reviewer for his constructive suggestions and comments.
1. Regarding table 3, we modified the title for clarity as all the included studies were Randomized Clinical Trials:
Laparoscopic versus open gastrectomy trials for GC
Randomized clinical trials on laparoscopic versus open gastrectomy for gastric cancer
2. The repetition was fixed as requested.